# The Association between Physical Activity and Intrinsic Capacity in Chinese Older Adults and Its Connection to Primary Care: China Health and Retirement Longitudinal Study (CHARLS)

**DOI:** 10.3390/ijerph20075361

**Published:** 2023-03-31

**Authors:** Mengping Zhou, Li Kuang, Nan Hu

**Affiliations:** 1Department of Medical Epidemiology and Biostatistics, Karolinska Institute, 17177 Stockholm, Sweden; 2Department of Health Administration, School of Public Health, Sun Yat-sen University, Guangzhou 510080, China; 3Department of Biostatistics, FIU Robert Stempel College of Public Health and Social Work, Miami, FL 33199, USA; 4Department of Family and Preventive Medicine, University of Utah School of Medicine, Salt Lake City, UT 84132, USA

**Keywords:** physical activity, intrinsic capacity, primary care, older adults, cognitive function

## Abstract

Background: In 2015, intrinsic capacity (IC) was proposed by the WHO as a new measure for healthy aging. Evidence has shown that physical activity (PA) benefits the physical and mental health of older adults. However, the association between PA and IC among older adults was not well evaluated or reported. This study aims to investigate the association between PA and general and specific IC among Chinese older adults. Method: The study included individuals aged 60 and above from the China Health and Retirement Longitudinal Study in 2015. The IC scores were constructed based on the WHO concept of five domains: psychological capacity, cognition, locomotion, vitality, and sensory abilities. Total PA and leisure PA were measured based on different activity purposes. Linear mixed-effects models and generalized linear mixed-effects models were developed to assess the associations between PA and IC. Results: A total of 3359 participants were included in this study. Older adults who reported some PA were associated with a higher composite IC score, with a mean difference of 0.14 (95% CI: 0.09–0.18, *p* < 0.001) compared to those who reported no PA. In terms of leisure PA, physically active adults had a higher composite IC score with a mean difference of 0.06 (95% CI: 0.03–0.09, *p* < 0.001). Older adults with a high level of leisure PA also had a significantly higher composite IC score (diff. in mean = 0.07, 95% CI: 0.01–0.13, *p* < 0.05) compared to those with low-level leisure PA. In addition, PA was positively and significantly associated with three specific IC domains: locomotion, cognition, and vitality. Conclusions: Improving both general and leisure PA can be an effective way to prevent the decline in IC among older adults, thus reducing the personal and public load of primary healthcare for aging countries such as China.

## 1. Introduction

According to a recent United Nations (UN) report, the proportion of adults 65 years and above is projected to increase from 9% in the year 2019 to 16% in the year 2050 [1]. This proportion was already at 13.5% in 2020 in China [2]. To address this rapidly aging global population, in 2015, the World Health Organization (WHO) proposed the idea of “Healthy Aging” in its “Global Report on Aging and Health”. This report emphasized developing and maintaining the functional capacity to enable older age wellbeing [3]. Functional ability is determined by intrinsic capacity (IC) and the relevant environment with which an individual engages as well as the interactions between the two. IC is defined as the composite of all the physical and mental capacities of an individual and includes five pivotal domains: locomotion, vitality, cognition, psychological capacities, and sensory abilities [4].

The decline in IC is prevalent among older adults. A recent population-based cohort study showed that two-thirds to three-quarters of adults 65 years and above experienced declines in one or more domains of IC [5]. The decline in IC was shown to increase the risk of dependence, falls, and mortality in community-dwelling older adults [5,6,7] and to increase the chance of nursing home stays among nursing home residents [8]. In addition, a higher IC score was found to be associated with reduced risks in 1-year mortality and functional dependency for hospitalized Chinese older adults [9] and was reported to promote the mental and physical health-related quality of life (QOL) among older adults in New Zealand [10]. These results indicated that maintaining the stability of IC among older adults played a crucial role in maintaining their functional abilities and helped to avoid or delay negative health outcomes in different clinical settings.

As the main care provider for older adults, primary care providers (PCPs) play a vital role in helping to prevent them from declining in IC both physically and psychosocially. To this end, the WHO issued the Integrated Care for Older People guidelines and handbook in both 2017 [11] and 2019 [12], which recommended simple interventions for the management and care of decline in different IC domains for older adults under primary care settings. Among these interventions, physical activity (PA) was strongly recommended, although the evidence for its efficacy among older adults was considered moderate at that time. Based on the theory of multisystemic benefits (e.g., endocrine, neuromuscular, metabolic, and cardiorespiratory) against age-related deterioration, lifetime PA may help to attenuate the loss of many important biological properties affected by aging, such as functional ability [13]. The Copenhagen Consensus statement 2019 expressed that physically active older adults show benefits in their physical and cognitive functions (such as IC, mobility, psychological wellbeing, and QOL) compared with physically inactive older adults [14].

Although the above studies demonstrated some effects of PA on IC, the studies were not based on quantitative research. So far, only a single-blinded randomized controlled trial (RCT) in Japan reported that both aerobic training and resistance training had a short-term benefit on IC among older adults [15]. However, this study only included subjects with subjective memory concerns, so it can hardly be generalized to the general population of older adults. As such, there is still a lack of knowledge about how PA can impact IC and which domains of IC may be affected.

To the best of our knowledge, there has been no study that directly examined the association between PA and IC among the general population of Chinese older adults. Leveraging a large sample longitudinal health survey in China, namely, the China Health and Retirement Longitudinal Study (CHARLS), we conducted this research to investigate the association between PA and general IC and specific IC domains among Chinese older adults. Our study aims to fill this knowledge gap and potentially help scholars to further understand the role of PA on IC among older adults in China.

## 2. Methods

### 2.1. Study Design

In this study, we used the CHARLS survey data collected in 2015, since the research team did not collect the Biomarker questionnaire in the latest 2018 survey data. CHARLS is a nationally representative longitudinal cohort study of individuals aged 45 years old and above in China. Its baseline survey was conducted in 2011–2012, and a further follow-up survey was conducted every two years. Based on a four-stage stratified cluster sampling method, CHARLS selected participants in 450 communities of 150 county-level units from 28 provinces in China. Detailed information about the purpose, design, sample, and questionnaires of the CHARLS is available in other articles [16]. The CHALRS research team has obtained ethical approval from the institutional review board at Peking University Health Science Center. The ethical approval number was IRB00001052-11015.

Among the 21,095 survey participants in 2015, we excluded a total of 8149 subjects under the age of 60 years, a total of 4855 individuals who had no information in the Biomarker questionnaire or the Health Status and Functioning questionnaire, a total of 4238 individuals who were not sampled for PA questions, and a total of 494 individuals missing information on educational status, self-reported health, and activities of daily life (ADL). Finally, 3359 participants were included in this cross-sectional study. A flowchart of the data procedure is presented in Figure 1.

### 2.2. Measurement of Intrinsic Capacity

The measurement of IC was based on a recently published paper that validated the IC using CHARLS data [17]. Locomotion was assessed using the Static Balance Test (semi-tandem stand, full-tandem stand, and side-by-side stand), 2.5 m walking speed, and chair-stand test. Since the Static Balance Test was measured by points (0–4), the walking and chair-stand time was also divided into four scores based on the five quartiles, and the total score was summed for these three test scores, ranging from 0 to 12 (the higher, the better locomotion). Cognition was measured by an adapted Chinese version of the Mini-Mental Status Examination (MMSE), which tests 4 aspects of cognitive abilities: orientation (recognition of today’s date, day of the week, and current season, 0 to 5 score); memory (immediate and 5 min delayed recall of a list of 10 Chinese nouns, 0 to 10 score); calculation (test of serial subtractions of 7 from 100, 0 to 5 score); and visuoconstruction (reproducing a picture of two overlapped pentagons, 0 to 1 score). The total cognitive function score varied from 0 to 31, with higher values meaning better cognitive function. Psychological capacity was measured by depressive symptoms using the Center for Epidemiologic Studies Depression Scale (CES-D-10). The response scale for the CES-D-10 includes 10 questions regarding how the participant felt and behaved during the past week, with the total score ranging from 0 to 30. A higher score indicates a higher level of depressive symptoms, and a cut-off score of ≥10 was the borderline for depression [18]. Sensory capacity was measured by self-report hearing and vision status. Participants were asked to rate their hearing, eyesight at a distance, and eyesight up close as excellent, very good, good, fair, or poor, corresponding to a score of 4, 3, 2, 1, and 0, respectively. The participant was treated as having sensory impairments if any of the statuses were rated as poor. The eyesight score was the average of the eyesight score at a distance and the eyesight score up close. The sensory score was the sum of the hearing score and eyesight score, ranging from 0 to 8. Vitality was measured by the handgrip strength and Forced Expiratory Volume (FEV), using a hand-held dynamometer and spirometer separately. Handgrip strength was measured as an average of two measurements of the dominant hand (if both hands were reported as the dominant hand, we took the average of the larger measure). Three technically satisfactory blows of FEV were recorded, and the highest was used in the analysis. A vitality Z-score was established by taking the average of handgrip strength Z-scores and FEV Z-scores. Finally, the composite IC Z-score (the mean of the locomotion Z-score, cognition Z-score, sensory Z-score, psychological Z-score, and vitality Z-score) was used as our study outcome for the general IC.

### 2.3. Measurement of Physical Activity

The CHARLS study fielded a localized short version of the globally recognized International Physical Activity Questionnaire (IPAQ), which measured the frequency and duration of intensive-, moderate- and light-intensity PA. Two variables of PA were generated to represent both leisure physical activity (LPA, aims for exercise and entertainment only) and total physical activity (TPA, aims for exercise, entertainment, job demand, and other purposes). The responses on daily PA duration for each PA type were coded as 1 ( ≤0.5 h), 2 (between 0.5 and 2 h), 3 (between 2 and 4 h), and 4 (≥4 h). The weekly PA duration score was calculated by multiplying the frequency and the daily PA duration index for each activity type. Subsequently, the PA scores were calculated using metabolic equivalent (MET) multipliers [19] as follows: (1) LPA score = 8.0 × leisure vigorous activity weekly duration score + 4.0 × leisure moderate activity weekly duration score + 3.3 × leisure walking weekly duration score and (2) TPA score = 8.0 × total vigorous activity weekly duration score + 4.0 × total moderate activity weekly duration score + 3.3 × total walking weekly duration score. We then separated the TPA/LPA score into two groups with scores equal to 0 and higher than 0, indicating participants who engage in PA and those who do not. For those who reported some PA, we further divided them into low, moderate, and high PA groups based on three quartiles of the TPA/LPA score.

### 2.4. Confounding Variable

Selection of confounding variables was primarily based on the literature. Confounders were selected if they were considered as correlated with both PA and IC and not intermediators for the association between PA and IC. The selection was subject to the availability of the 2015 CHARLS data. Specially, we included demographic features (age, sex, marital status, and education), the availability and characteristics of health resources (current residence, GDP per capita (PGDP) at prefecture-city-level, and the economic region at province-level), health status (self-reported health, ADLs limitations, instrumental activities of daily life (IADLs) limitations, number of chronic diseases, number of disabilities).

The definitions and assignments of all variables are shown in Table A1 (Appendix A). The association between PA and IC and potential confounders is shown in Figure 2.

### 2.5. Statistical Analysis

Characteristics of the study population were summarized as frequency (N) and percentage (%) for categorical variables and as mean ± standard deviation (SD) for continuous variables with normality and approximate normality. Chi-squared tests, a two-sample Student’s *t*-test, and one-way analysis of variance (ANOVA) were used to test the differences in the covariates among the different PA levels.

The associations between PA and IC were investigated in two different ways. In the first method, we examined whether engaging in PA or not (yes/no) was associated with IC and 5 IC domains. In the second method, we investigated whether different TPA/LPA levels (low/moderate/high) were associated with IC and each of the 5 IC domains by excluding participants who reported no PA. A linear mixed-effects model (LMEM) was used for assessing the associations between PA levels and continuous IC outcomes. A generalized linear mixed-effects model (GLMEM) with a logit link function was used for binary IC outcomes. All models were adjusted for the covariates listed in Section 2.4. The regression coefficient (β), odds ratio (OR), 95% confidence interval (CI), and *p*-value were reported.

For continuous outcomes, the LMEM was presented with the following mathematical equation:(1)$$Y_{i,\;j,\;k}\;=\beta_{0}+\beta_{1}\;X_{i,\;j,\;k}\;+\sum_{n=2}^{m}\beta_{n}X_{n,\;i,\;j,\;k}+u_{0,\;j}\;+\;u_{0,\;j,\;k}\;+e_{i,\;j\;,k}$$

For binary outcomes, the GLMEM was presented with the following mathematical equation:
(2)$$E\;(Y_{i,\;j\;,k})\;=invlogit\;(\beta_{0}\;+\beta_{1}\;X_{i\;,j\;,k}\;+\sum_{n=2}^{m}\beta_{n}X_{n,\;i,j,k}+u_{0,\;j}\;+\;u_{0,\;j,k}\;+e_{i,\;j,k})$$

The subscript *k* is for the county (*k* = 1…*k*), *j* is for the household (*j* = 1…*j*), and the subscript *i* is for individual pupils (*i* = 1, 2). The *u*-terms *u_0,j_* in equations 1 and 2 are random residual error terms at the household level, and *u_0,j,k_* are random residual error terms at the county level. *e_i_*_,*j*,*k*_ is the residual error at the individual level. The regression coefficients βs are referred to as the fixed-effect coefficients and are not assumed to vary across households and counties.

All statistical analyses were performed using R version 4.2.0. All statistical tests were two-sided, and *p*-values < 0.05 were considered statistically significant.

### 2.6. Reporting

Reporting of the study findings followed the guidelines of strengthening the reporting of observational studies in epidemiology (STROBE).

## 3. Results

### 3.1. Subjects’ Characteristics

Table 1 shows the descriptive statistics of all covariates for the total sample and for each TPA/LPA group. Among the 3359 subjects, 472 reported no, 1137 reported low-, 787 reported moderate-, and 963 reported high-level TPA. In addition, 2045 participants reported no, 757 reported a low level, 238 reported a moderate level, and 319 reported a high level of LPA. The older adults reporting higher TPA levels were more likely to be male, had better self-reported health and less ADL and IADL limitation, and tended to live in the Western region of China. In addition, study subjects reporting higher levels of LPA tended to have less IADL limitation and live in places with greater PGDP.

### 3.2. Summary of Intrinsic Capacity

Table 2 reports the summary statistics of IC and IC domains for the total sample and for each TPA and LPA level. Among the 3359 subjects, 1239 (37%) had depression, and 2540 (75.8%) reported sensory impairment. The mean of the cognition score, vitality Z-score, locomotion score, and composite IC Z-score was 10.37 ± 5.16, −0.29 ± 0.81, 7.04 ± 2.68, and −0.07 ± 0.63, respectively. All five IC domains and the composite IC score were significantly different between participants with no TPA and those with some TPA, whereas the difference in the depression proportion and cognition score was not statistically significant when comparing across different TPA levels. As for LPA (yes/no), we observed significant associations with depression, cognition score, and composite IC Z-score. When further comparing across different LPA levels, the difference in sensory impairment, vitality Z-score, and locomotion score became statistically significant.

### 3.3. Association between Physical Activity and Intrinsic Capacity among Chinese Older Adults

Table 3 presents all the estimated regression coefficients for associations between TPA/LPA and all IC outcomes. Older adults who reported some TPA had a mean composite IC score of 0.14 (95% CI: 0.09–0.18, *p* < 0.001) higher than those without TPA. Furthermore, among adults who reported TPA, those with a high or moderate TPA level also had a higher composite IC score, but the association was not significant. For the specific domain, significant associations were observed between TPA and locomotion (diff. in mean locomotion score between “yes” and “no” = 0.56, 95% CI: 0.34–0.79, *p* < 0.001; diff. in mean locomotion score between “moderate” and “low” = 0.36, 95% CI: 0.16–0.56, *p* < 0.001; diff. in mean locomotion score between “high” and “low” = 0.55, 95% CI: 0.35–0.75, *p* < 0.001), between TPA and vitality score (diff. in mean vitality score between “yes” and “no” = 0.12, 95% CI: 0.07–0.18, *p* < 0.001; diff. in mean vitality score between “moderate” and “low” = 0.07, 95% CI: 0.02–0.12, *p* < 0.01; diff. in mean vitality score between “high” and “low” = 0.09, 95% CI: 0.04–0.14, *p* < 0.001), and between TPA and cognition score (diff. in mean cognition score between “yes” and “no” = 1.62, 95% CI: 1.20–2.04, *p* < 0.001). However, older adults who reported a high or moderate TPA level had a higher risk of depression, compared to those who reported a low level. No significant relationship was found between TPA level and sensory impairment.

As for LPA, those who reported some LPA had a significantly higher mean of composite IC score compared to adults without LPA (diff. in mean = 0.06, 95% CI: 0.03–0.09, *p* < 0.001). Among adults with some LPA, those with a high level of LPA had a mean composite IC score significantly higher than those with a low level of LPA (diff. in mean = 0.07, 95% CI: 0.01–0.13, *p* < 0.05). For the IC domains, older adults with a high LPA level had a higher locomotion score in the amount of 0.38 (95% CI: 0.09–0.67, *p* < 0.05) and a vitality score of 0.13 (95% CI: 0.05–0.21, *p* < 0.001) compared to a low level. Compared to adults who reported no LPA, those with some LPA had a higher cognition score in the amount of 0.92 (95% CI: 0.62–1.23, *p* < 0.001) and a lower risk for depression (OR = 0.75, 95% CI: 0.75–0.76, *p* < 0.001), whereas these associations were not observed when comparing across different LPA levels. Consistent with TPA, no significant association was found between LPA and sensory impairment.

Figure 3 and Figure 4 plot the estimated IC score and IC domain versus participants’ TPA levels and LPA levels, respectively. In both plots, a zero score on TPA/LPA served as a reference group.

## 4. Discussion

By using the Chinese nationwide CHARLS survey data, our research found that PA levels are positively associated with the composite IC score and three IC domains (locomotion, cognition, and vitality) among older adults. Our study indicated that promoting older adults’ PA in their everyday life may be an effective approach to prevent a decline in their IC. To the best of our knowledge, this is the first study demonstrating the relationship between PA and IC (and IC domains) among older adults in China.

We observed consistent patterns for the associations between PA and composite IC index when using different PA measurements, that is, older adults who reported some TPA/LPA had an increased IC score, compared to those without TPA/LPA. Furthermore, older adults who reported a high or moderate PA level had an increased IC score compared to a low level, although this association was only significant for LPA. Our results found the positive association between PA and IC were consistent with previous studies [15,20,21], further confirming the potential beneficial effects of PA among older adults. Growing evidence also identified PA as a potential low-cost intervention in reducing mortality [22], falls [23], and improving the quality of life [24] for older adults. Compared with these health outcomes, the composite IC with continuous scoring potentially captures and quantifies the trajectory of healthy aging. Our results confirm the promising effects of PA on this composite healthy aging marker, namely, IC, which could facilitate exercise promotion and serve as evidence for monitoring older adults.

More specifically, the associations between PA and IC were mainly reflected in three domains: locomotion, cognition, and vitality, and the study in Japan found the significant associations in the same domains but not the cognition [15]. Supervised physical exercise programs addressed towards older adults have been shown to contribute to an improvement in physical parameters such as cardiorespiratory fitness, gait, muscle strength, and clinical balance outcomes [25,26,27], which in turn delays several geriatric syndromes and improves locomotion [28] and vitality [29,30]. Besides these two domains of IC, PA was also identified as a potent lifestyle factor critical in reducing cognitive decline and improving cognitive function in older adults with different study designs [31,32,33,34]. However, we also observed minor differences in the associations with some domains between TPA and LPA. For example, only participants with a high LPA level had an increased locomotion and vitality score (compared to a low PA level), whereas positive associations were found for each TPA level. In general, the estimator was larger for the association with TPA than with LPA, which indicates that PA due to no-leisure purposes also works for improving some IC domains.

We did not find significant associations between either TPA or LPA and sensory impairment. Although there were studies that reported a positive relationship between PA and sensory processing in children [35], few studies have investigated the association in older adults. Among older adults, lifestyle interventions may not readily slow down this impairment process, since the decline in sensory ability is significantly related to age, and there is a common underlying factor that drives the age-related deterioration of sensory processes [36]. We did not find consistent relationships between depression and LPA or TPA. Compared to those with a low TPA level, older adults with a moderate or high TPA level are more likely to experience depression. In contrast, older adults who reported LPA are less likely to experience depression compared to those who reported no LPA. Based on the synthesized data, a recent meta-analysis focusing on the associations between PA and depression reported significant mental health benefits from being physically active even if the level of PA was below the public health recommendation levels [37]. Likewise, increased aerobic exercise or strength training has been shown to reduce depressive symptoms significantly [38]. Although many studies showed the beneficial effects of PA in reducing the risk of depression, there was also evidence of no antidepressant effects of habitual PA [38] and self-reported PA [39]. The positive association we found between depression and TPA may be due to the way depression is measured and the criteria used to define it. Depression was self-reported in the CHARLS questionnaires, and the classification rule (threshold value) based on the original depression score may lead to a detection of depressive symptoms instead of a clinical diagnosis for depression. The classification using the CHARLS threshold may overestimate the number of subjects with depression and then generate biased results [40]. In addition, the opposite association between depression with TPA and LPA somewhat indicates that the effect of PA on mental health is dependent on the purpose of PA. Specifically, PA for exercise and entertainment helps to promote mental health, whereas PA for job demand does not. The discrepant results demonstrate that future investigation is required to address this discrepancy among older adults.

We adopted a random intercept model (at the levels of household and county), as we observed obvious differences in intercepts across counties and households instead of a difference in the slope. In addition, we conducted chi-squared tests to compare the random intercept model (at both the household and county level) with both the random intercept (at both household and county level) and random slope (at county level) model. There were no significant differences between the two types of models for most analytical results. In terms of model fitting, the random intercept model has better model fitting, as evidenced by a smaller value of both the Akaike information criterion (AIC) and the Bayesian information criterion (BIC). We did not consider a household-specific random slope model, since the sample size of the household (2363) and the average observation per household (1.42) were not supportive for random slopes at the household level.

This study has several limitations. First, some potential confounding factors for the relationship between PA and IC were not available in the CHARLS data, such as the study subjects’ employment status and occupation (if employed). Second, we only used cross-sectional data from the year 2015, since limited participants finished all four waves of the survey data. Future studies can use further prospective cohort studies to evaluate the effect of PA and to further explore the most appropriate exercise type and length. Third, our study used cross-sectional and observational data. For this reason, this study cannot identify any causal relationship between PA and IC.

## 5. Conclusions

Our study underlines positive associations between PA on IC as well as three domains of IC (locomotion, cognition, and vitality) in Chinese older adults. Thus, we highly recommend investigating an optimal physical training protocol that is suitable for older adults in China. In addition, we believe that raising awareness of the importance of engaging in regular PA among older adults will benefit this population by preventing a decline in their IC, especially their cognitive functions. We suggest that PA be prescribed with a progressive individualized plan, just like other medical treatments for older adults in primary care settings. For example, the promotion of low-intensity exercises, such as Tai Chi and Yoga, walking, and guided group fitness classes for senior citizens, can be an effective move to improve IC among older adults in China. These will potentially reduce the healthcare load and cost of older retirees, primary care, and long-term care facilities in aging countries. As our study did not find statistically significant associations between PA and sensory impairment and depression among older adults in China, we suggest that more studies in the future examine these relationships to provide more evidence of the role of PA in these important mental health outcomes.

## Figures and Tables

**Figure 1 ijerph-20-05361-f001:**
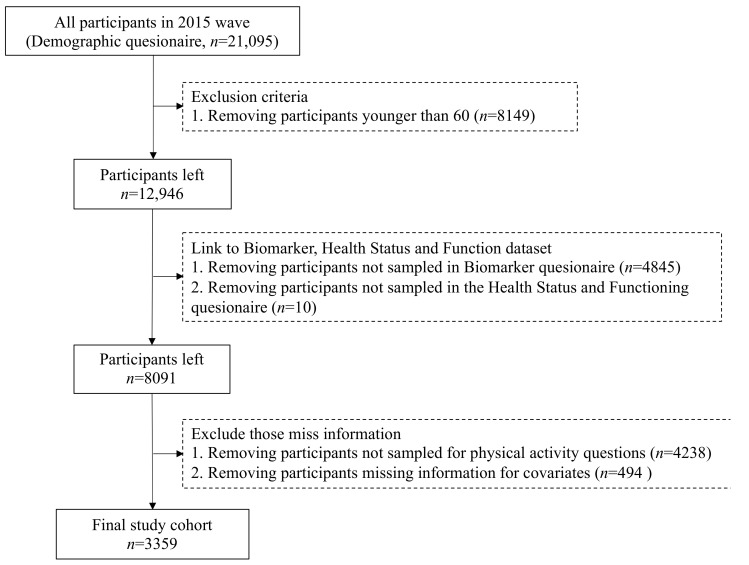
Data extraction and management flowchart.

**Figure 2 ijerph-20-05361-f002:**
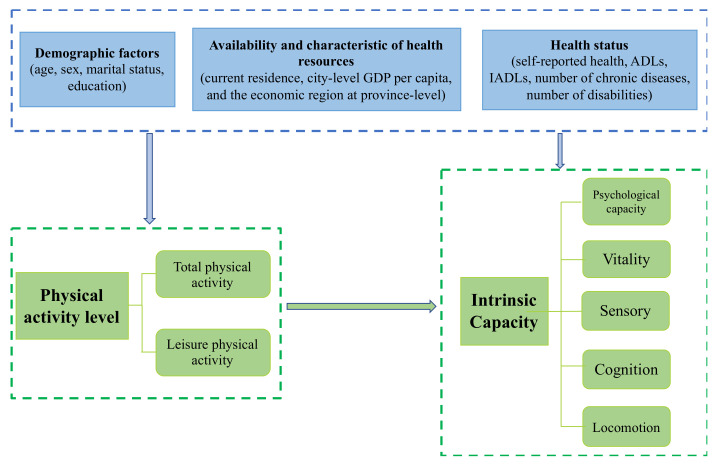
Diagram of association between PA and IC, including potential confounders.

**Figure 3 ijerph-20-05361-f003:**
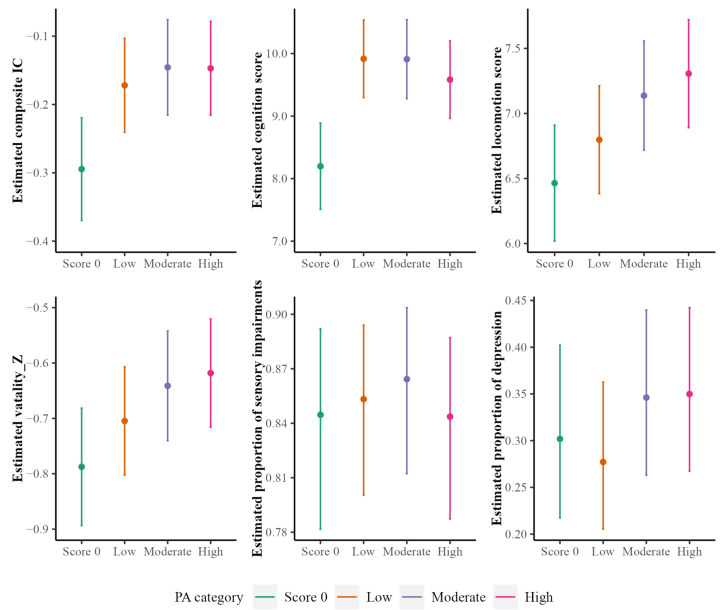
Estimated general IC scores (and scores/proportion for each of the five IC domains) with 95% confidence intervals (CIs) versus TPA levels. The estimation is based on a linear mixed-effects model for continuous outcomes and a generalized linear mixed-effects model for binary outcomes.

**Figure 4 ijerph-20-05361-f004:**
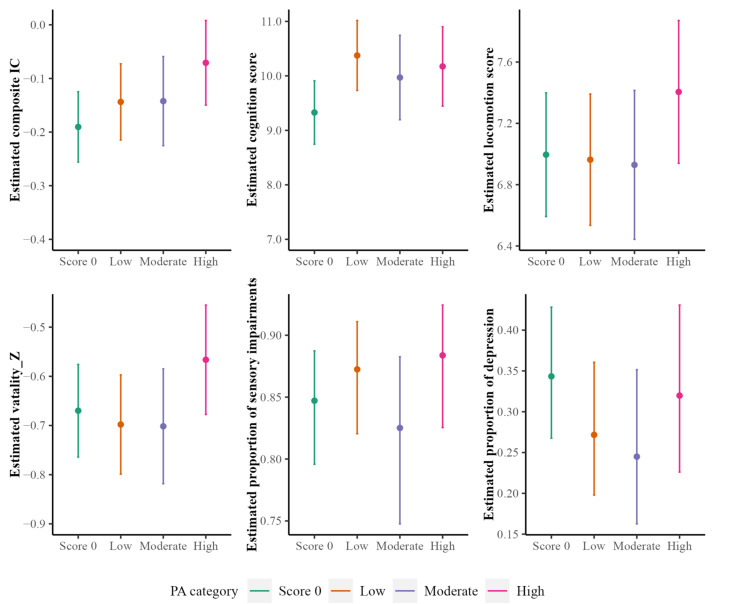
Estimated general IC scores (and scores/proportion for each of the five IC domains) with 95% confidence intervals (CIs) versus LPA levels. The estimation is based on a linear mixed-effects model for continuous outcomes and a generalized linear mixed-effects model for binary outcomes.

**Table 1 ijerph-20-05361-t001:** Characteristics of study participants.

Variable		All Subjects (*n* = 3359)	TPA Score (*n* = 3359)			TPA Score > 0 (*n* = 2887)				LPA Score (*n* = 3359)			LPA Score > 0 (*n* = 1314)			
			Score = 0 (*n* = 472)	Score > 0 (*n* = 2887)	*p*	Low (*n* = 1137)	Moderate (*n* = 787)	High (n = 963)	*p*	Score = 0 (*n* = 2045)	Score > 0 (*n* = 1314)	*p*	Low (*n* = 757)	Moderate (*n* = 238)	High (*n* = 319)	*p*
Age, Mean (SD)		68.11 (6.56)	70.27 (7.67)	67.76 (6.30)	<0.001	69.23 (6.79)	67.71 (6.27)	66.07 (5.18)	<0.001	67.52 (6.39)	69.03 (6.72)	<0.001	69.31 (6.87)	68.50 (6.34)	68.75 (6.63)	0.271
Sex, *n* (%)	Male	1638 (48.8)	208 (44.1)	1430 (49.5)	0.028	530 (46.6)	375 (47.6)	525 (54.5)	<0.001	971 (47.5)	667 (50.8)	0.064	381 (50.3)	112 (47.1)	174 (54.5)	0.203
	Female	1721 (51.2)	264 (55.9)	1457 (50.5)		607 (53.4)	412 (52.4)	438 (45.5)		1074 (52.5)	647 (49.2)		376 (49.7)	126 (52.9)	145 (45.5)	
Marital status, *n* (%)	Without spouse	646 (19.2)	126 (26.7)	520 (18.0)	<0.001	238 (20.9)	156 (19.8)	126 (13.1)	<0.001	359 (17.6)	287 (21.8)	0.002	165 (21.8)	51 (21.4)	71 (22.3)	0.972
	With spouse	2713 (80.8)	346 (73.3)	2367 (82.0)		899 (79.1)	631 (80.2)	837 (86.9)		1686 (82.4)	1027 (78.2)		592 (78.2)	187 (78.6)	248 (77.7)	
Education, *n* (%)	Elementary school and below	2723 (81.1)	412 (87.3)	2311 (80.0)	<0.001	877 (77.1)	638 (81.1)	796 (82.7)	0.018	1732 (84.7)	991 (75.4)	<0.001	574 (75.8)	190 (79.8)	227 (71.2)	0.092
	Secondary school	598 (17.8)	59 (12.5)	539 (18.7)		243 (21.4)	137 (17.4)	159 (16.5)		306 (15.0)	292 (22.2)		169 (22.3)	43 (18.1)	80 (25.1)	
	College and above	38 (1.1)	1 (0.2)	37 (1.3)		17 (1.5)	12 (1.5)	8 (0.8)		7 (0.3)	31 (2.4)		14 (1.8)	5 (2.1)	12 (3.8)	
Self-reported health status, *n* (%)	Poor	1043 (31.1)	201 (42.6)	842 (29.2)	<0.001	374 (32.9)	205 (26.0)	263 (27.3)	0.002	661 (32.3)	382 (29.1)	0.126	234 (30.9)	70 (29.4)	78 (24.5)	0.057
	Fair	1671 (49.7)	197 (41.7)	1474 (51.1)		565 (49.7)	423 (53.7)	486 (50.5)		994 (48.6)	677 (51.5)		394 (52.0)	118 (49.6)	165 (51.7)	
	Good	645 (19.2)	74 (15.7)	571 (19.8)		198 (17.4)	159 (20.2)	214 (22.2)		390 (19.1)	255 (19.4)		129 (17.0)	50 (21.0)	76 (23.8)	
ADL limitations, *n* (%)	No	2473 (73.6)	285 (60.4)	2188 (75.8)	<0.001	817 (71.9)	623 (79.2)	748 (77.7)	<0.001	1478 (72.3)	995 (75.7)	0.027	560 (74.0)	178 (74.8)	257 (80.6)	0.066
	Yes	886 (26.4)	187 (39.6)	699 (24.2)		320 (28.1)	164 (20.8)	215 (22.3)		567 (27.7)	319 (24.3)		197 (26.0)	60 (25.2)	62 (19.4)	
IADL limitations, *n* (%)	No	2163 (64.4)	235 (49.8)	1928 (66.8)	<0.001	693 (60.9)	565 (71.8)	670 (69.6)	<0.001	1289 (63.0)	874 (66.5)	0.040	485 (64.1)	156 (65.5)	233 (73.0)	0.016
	Yes	1196 (35.6)	237 (50.2)	959 (33.2)		444 (39.1)	222 (28.2)	293 (30.4)		756 (37.0)	440 (33.5)		272 (35.9)	82 (34.5)	86 (27.0)	
Number of chronic diseases, *n* (%)	0	672 (20.0)	79 (16.7)	593 (20.5)	0.117	208 (18.3)	164 (20.8)	221 (22.9)	0.099	441 (21.6)	231 (17.6)	<0.001	132 (17.4)	42 (17.6)	57 (17.9)	0.642
	1	864 (25.7)	133 (28.2)	731 (25.3)		290 (25.5)	195 (24.8)	246 (25.5)		554 (27.1)	310 (23.6)		171 (22.6)	65 (27.3)	74 (23.2)	
	≥2	1823 (54.3)	260 (55.1)	1563 (54.1)		639 (56.2)	428 (54.4)	496 (51.5)		1050 (51.3)	773 (58.8)		454 (60.0)	131 (55.0)	188 (58.9)	
Number of disablities, *n* (%)	0	2760 (82.2)	360 (76.3)	2400 (83.1)	<0.001	937 (82.4)	662 (84.1)	801 (83.2)	0.140	1681 (82.2)	1079 (82.1)	0.663	623 (82.3)	189 (79.4)	267 (83.7)	0.172
	1	481 (14.3)	82 (17.4)	399 (13.8)		156 (13.7)	111 (14.1)	132 (13.7)		288 (14.1)	193 (14.7)		105 (13.9)	40 (16.8)	48 (15.0)	
	≥2	118 (3.5)	30 (6.4)	88 (3.0)		44 (3.9)	14 (1.8)	30 (3.1)		76 (3.7)	42 (3.2)		29 (3.8)	9 (3.8)	4 (1.3)	
Economic region, *n* (%)	West	965 (28.7)	95 (20.1)	870 (30.1)	<0.001	271 (23.8)	249 (31.6)	350 (36.3)	<0.001	606 (29.6)	359 (27.3)	0.305	193 (25.5)	62 (26.1)	104 (32.6)	0.053
	Middle	1245 (37.1)	187 (39.6)	1058 (36.6)		443 (39.0)	285 (36.2)	330 (34.3)		742 (36.3)	503 (38.3)		283 (37.4)	98 (41.2)	122 (38.2)	
	East	1149 (34.2)	190 (40.3)	959 (33.2)		423 (37.2)	253 (32.1)	283 (29.4)		697 (34.1)	452 (34.4)		281 (37.1)	78 (32.8)	93 (29.2)	
PGDP, 1000 yuan, Mean (SD)		46.20 (27.74)	45.79 (27.85)	46.26 (27.73)	0.668	48.64 (28.15)	47.32 (27.81)	42.59 (26.78)	<0.001	44.08 (27.20)	49.49 (28.25)	<0.001	50.55 (28.00)	49.07 (28.16)	47.29 (28.88)	0.024
Current residence, n (%)	Rural	2216 (66.0)	330 (69.9)	1886 (65.3)	0.051	648 (57.0)	479 (60.9)	759 (78.8)	<0.001	1517 (74.2)	699 (53.2)	<0.001	412 (54.4)	125 (52.5)	162 (50.8)	0.536
	Urban	1143 (34.0)	142 (30.1)	1001 (34.7)		489 (43.0)	308 (39.1)	204 (21.2)		528 (25.8)	615 (46.8)		345 (45.6)	113 (47.5)	157 (49.2)	

Note: TPA = total physical activity, LPA = leisure physical activity, ADL = activities of daily life, IADL = instrumental activities of daily life, PGDP = gross domestic product (GDP) per capital at prefecture city level.

**Table 2 ijerph-20-05361-t002:** Summary of IC and IC domains by different TPA/LPA levels.

Intrinsic Capacity		All Subjects (*n* = 3359)	TPA Score (*n* = 3359)			TPA Score > 0 (*n* = 2887)				LPA Score (*n* = 3359)			LPA Score > 0 (*n* = 1314)			
			Score = 0 (*n* = 4,72)	Score > 0 (*n* = 2887)	*p*	Low (*n* = 1137)	Moderate (*n* = 787)	High (*n* = 963)	*p*	Score = 0 (*n* = 2045)	Score > 0 (*n* = 1314)	*p*	Low (*n* = 757)	Moderate (*n* = 238)	High (*n* = 319)	*p*
Depression, *n* (%) (11 missing)	No	2109 (63.0)	265 (56.5)	1844 (64.1)	0.002	743 (65.6)	506 (64.5)	595 (61.9)	0.197	1210 (59.3)	899 (68.7)	<0.001	513 (68.1)	166 (69.7)	220 (69.2)	0.874
	Yes	1239 (37.0)	204 (43.5)	1035 (35.9)		390 (34.4)	278 (35.5)	367 (38.1)		829 (40.7)	410 (31.3)		240 (31.9)	72 (30.3)	98 (30.8)	
Sensory impairments, *n* (%) (7 missing)	No	812 (24.2)	151 (32.3)	661 (22.9)	<0.001	287 (25.3)	159 (20.2)	215 (22.3)	0.028	514 (25.2)	298 (22.7)	0.102	173 (22.9)	66 (27.7)	59 (18.5)	0.036
	Yes	2540 (75.8)	317 (67.7)	2223 (77.1)		847 (74.7)	628 (79.8)	748 (77.7)		1526 (74.8)	1014 (77.3)		582 (77.1)	172 (72.3)	260 (81.5)	
Cognition score, Mean (SD)		10.37 (5.16)	7.89 (5.10)	10.78 (5.06)	<0.001	10.78 (5.17)	11.04 (5.12)	10.56 (4.87)	0.111	9.74 (5.07)	11.35 (5.16)	<0.001	11.38 (5.09)	10.87 (5.28)	11.65 (5.24)	0.143
Vitality_Z, Mean (SD)		−0.29 (0.81)	−0.57 (0.82)	−0.24 (0.80)	<0.001	−0.36 (0.81)	−0.23 (0.80)	−0.12 (0.76)	<0.001	−0.30 (0.80)	−0.28 (0.83)	0.777	−0.33 (0.83)	−0.34 (0.78)	−0.10 (0.85)	<0.001
Locomotion score, Mean (SD)		7.04 (2.68)	5.97 (2.98)	7.20 (2.59)	<0.001	6.64 (2.68)	7.34 (2.47)	7.74 (2.45)	<0.001	7.03 (2.69)	7.04 (2.68)	0.801	6.87 (2.70)	6.80 (2.64)	7.61 (2.56)	<0.001
Composite IC_Z, Mean (SD)		−0.07 (0.63)	−0.38 (0.65)	−0.02 (0.61)	<0.001	−0.09 (0.63)	0.02 (0.59)	0.03 (0.59)	<0.001	−0.12 (0.62)	0.002 (0.63)	<0.001	−0.04 (0.63)	−0.04 (0.62)	0.13 (0.62)	<0.001
TPA Score, Mean (SD)		112.95 (106.76)	0 (0)	131.41 (104.09)	<0.001	43.22 (17.97)	104.75 (21.41)	257.33 (78.13)	<0.001							
LPA Score, Mean (SD)		25.28 (44.48)								0 (0)	64.56 (50.12)	<0.001	36.50 (12.83)	65.90 (6.30)	130.16 (61.31)	<0.001

Note: TPA = total physical activity, LPA = leisure physical activity, CESD score >=10 indicates depression symptoms; sensory was impaired if hearing or eyesight was poor.

**Table 3 ijerph-20-05361-t003:** Associations between PA and IC among older adults in CHARLS 2015.

	Composite IC Z-Score β (95% CI)	Cognition Score β (95% CI)	Locomotion Score β (95% CI)	Vatality Z-Score β (95% CI)	Sensory Impairment OR (95% CI)	Depression OR (95% CI)
TPA (Yes/No) No (Score = 0)	Ref	Ref	Ref	Ref	Ref	Ref
Yes (Score > 0)	0.14 *** (0.09, 0.18)	1.62 *** (1.20, 2.04)	0.56 *** (0.34, 0.79)	0.12 *** (0.07, 0.18)	1.08 (0.84, 1.38)	1.04 (0.81, 1.34)
TPA (3 levels, when Score > 0)						
Low (Score: 43.22 ± 17.97)	Ref	Ref	Ref	Ref	Ref	Ref
Moderate (Score: 104.75 ± 21.41)	0.03 (−0.01, 0.07)	−0.01 (−0.40, 0.37)	0.36 *** (0.16, 0.56)	0.07 ** (0.02, 0.12)	1.08 (0.85, 1.37)	1.27 * (1.00, 1.6)
High (Score: 257.33 ± 78.13)	0.03 (−0.01, 0.06)	−0.34 (−0.73, 0.04)	0.55 *** (0.35, 0.75)	0.09 *** (0.04, 0.14)	0.93 (0.73, 1.17)	1.33 * (1.05, 1.68)
LPA (Yes/No) No (Score = 0)	Ref	Ref	Ref	Ref	Ref	Ref
Yes (Score > 0)	0.06 *** (0.03, 0.09)	0.92 *** (0.62, 1.23)	0.06 (−0.10, 0.23)	0.00 (−0.04, 0.04)	1.16 (0.96, 1.41)	0.75 *** (0.75, 0.76)
LPA (3 levels, when Score > 0)						
Low (Score: 36.50 ± 12.83)	Ref	Ref	Ref	Ref	Ref	Ref
Moderate (Score: 65.90 ± 6.30	0.00 (−0.07, 0.06)	−0.46 (−1.07, 0.16)	−0.08 (−0.4, 0.25)	−0.01 (−0.09, 0.08)	0.71 (0.49, 1.03)	0.85 (0.59, 1.23)
High (Score: 130.16 ± 61.31)	0.07 * (0.01, 0.13)	−0.25 (−0.81, 0.31)	0.38 * (0.09, 0.67)	0.13 *** (0.05, 0.21)	1.08 (0.76, 1.55)	1.15 (0.82, 1.61)

Note: * *p* < 0.05, ** *p* < 0.01, *** *p* < 0.001. Controlled for age, gender, educational level, marital status, self-reported health, the number of disabilities, the number of diseases, ADL limitation, IADL limitation, PGDP, ecoregion, and current residence.

## Data Availability

The datasets analyzed during the current study are publicly available at http://charls.pku.edu.cn (accessed on 22 March 2023).

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
