# Peer review of "The Association between Physical Activity and Intrinsic Capacity in Chinese Older Adults and Its Connection to Primary Care: China Health and Retirement Longitudinal Study (CHARLS)"

_ijerph, 2023, doi:10.3390/ijerph20075361_

Round 1

Reviewer 1 Report

Thank you for the opportunity to review this interesting manuscript titled “The Effect of Physical Activity on Intrinsic Capacity in Chinese Elderly Population and Its Implication to Primary Care: China Health and Retirement Longitudinal Study (CHARLS)”. This study aimed to examine the association between physical activity (PA) and intrinsic capacity (IC) in the general Chinese older population based on the CHARLS survey data. The paper provides relevant recommendations for engaging in regular PA among older adults to prevent decline in IC. Generally, the manuscript is well-written and clear, so my comments are relatively minor. Please see my specific comments below.

Abstract 

Please replace the term “elderlies” and “elderly” with “older adults” or “older people” throughout the manuscript.

Introduction

What is meant by the term “incident dependence”? Please define.

Discussion

Page 13 – “…since the decline of sensory is significantly age-related” Word missing?

Reviewer 2 Report

ijerph-2259646-review

Intrinsic capacity (IC), total physical activity (PA) and leisure PA were measured in 3,359 participants. Linear mixed-effects models and generalized linear mixed-effects models were developed to assess the associations between PA and IC. The Authors conclude that PA had positive and statistically significant effects on three specific IC domains: locomotion, cognition, and vitality.

Overall, it’s large, interesting analysis.

Comments

Association of PA to only three dimensions of IC should be discussed further. Perhaps the overall construction of IC should be revised? The lack of or even inverse association of PA to sensory impairments and depression may results from the fact that younger more active subjects report more often those problems.

Introduction: “…although the current evidence for the effect of PA among elderlies is moderate.” That’s not true. Benefits are well documented. You also describe examples in the discussion.

Avoid terms like “effects” or “increased the mean composite IC”. This is a cross-sectional study no firm cause-effect conclusions can be drawn.

The term “elderliest” is rather not used in the literature. Native English editor should be asked.

Several Editorial corrections are needed, e.g.:

`”Vatality scsore”

Reviewer 3 Report

The authors conducted an observational study to examine the cross-sectional association between physical activity and intrinsic capacity. By analyzing the data of 3359 community-dwelling old adults aged 60 years and older, the authors showed that higher levels of physical activity were, in general, associated with higher intrinsic capacity. Intrinsic capacity is an important issue. And this study attempts to address it from one perspective. Yet, there are some concerns.

  1. Introduction: “we conducted this research to investigate the effects of PA on general IC and specific IC domains among -.” According to this description, the authors aimed to investigate whether there is a causal relationship between physical activity and intrinsic capacity. However, a cross-sectional study was conducted, which precludes examining the temporal, let alone causal, relationship. A revision of the rationale is suggested.
  2. Methods (Study Design): Among the 21,095 survey participants in 2015, the authors included 3,359 participants in this study after excluding those with incomplete data. Please confirm that the numbers are correct.
  3. Methods (Covariates): A variety of covariates were considered in this study. However, it is unclear how these covariates were measured. A more detailed description is necessary.
  4. Methods (Covariates): “Covariates were considered based on the Direct Acyclic Graph (DAG).” However, figure 1 is not, by definition, DAG. In DAG, at least relationships among each covariate need to be specified. With such details, specific rules could then be applied to select the set of covariates for the adjustment. According to the description of the manuscript, the authors selected covariates based on whether they are common causes of physical activity levels and intrinsic capacity rather than based on DAG. As such, it is recommended that the authors state this instead of describing that DAG was applied. A revision of the description is strongly suggested.
  5. Methods (Statistical analysis): The authors applied a random-intercept (at the levels of household and county) model. However, the reasons are unclear. Moreover, why would the authors not consider random-coefficient (ransom intercepts and random slopes) models by, for instance, including an additional county-specific random slope for physical activity?
  6. Discussion: A major limitation of this study worth discussing is its cross-sectional nature, which precludes examining the temporal, let alone causal, relationship between physical activity and intrinsic capacity. A major related problem of this study is that reverse causality is possible.
  1. This study yields, at best, estimates of association measures, which could be quite different from effect measures. Please avoid using the following terms throughout the manuscript: effect, effective, etc. A more appropriate term would be “association” (or “relationship”). 

Round 2

Reviewer 2 Report

-

Reviewer 3 Report

All the issues have been addressed appropriately by the authors in this manuscript version.